# A Contact-Free Optical Device for the Detection of Pulmonary Congestion—A Pilot Study

**DOI:** 10.3390/bios12100833

**Published:** 2022-10-06

**Authors:** Ilan Merdler, Aviram Hochstadt, Eihab Ghantous, Lior Lupu, Ariel Borohovitz, David Zahler, Philippe Taieb, Ben Sadeh, Zeev Zalevsky, Javier Garcia-Monreal, Michael Shergei, Maxim Shatsky, Yoav Beck, Sagi Polani, Yaron Arbel

**Affiliations:** 1Department of Cardiology, Tel Aviv Medical Center, 6 Weizmann Street, Tel Aviv, Tel-Aviv University, Tel Aviv 69978, Israel; 2Donisi Health, Formerly Contin Use Biometrics Ltd., Tel Aviv 69978, Israel; 3Faculty of Engineering, Bar-Ilan University, Ramat Gan 52900, Israel; 4Department of Optics, University of Valencia, 46003 Valencia, Spain

**Keywords:** contact-free, increased lung fluid content, congestion, heart failure, camera, remote monitoring

## Abstract

Background: The cost of heart failure hospitalizations in the US alone is over USD 10 billion per year. Over 4 million Americans are hospitalized every year due to heart failure (HF), with a median length of stay of 4 days and an in-hospital mortality rate that exceeds 5%. Hospitalizations of patients with HF can be prevented by early detection of lung congestion. Our study assessed a new contact-free optical medical device used for the early detection of lung congestion. Methods: The Gili system is an FDA-cleared device used for measuring chest motion vibration data. Lung congestion in the study was assessed clinically and verified via two cardiologists. An algorithm was developed using machine learning techniques, and cross-validation of the findings was performed to estimate the accuracy of the algorithm. Results: A total of 227 patients were recruited (101 cases vs. 126 controls). The sensitivity and specificity for the device in our study were 0.91 (95% CI: 0.86–0.93) and 0.91 (95% CI: 0.87–0.94), respectively. In all instances, the observed estimates of PPVs and NPVs were at least 0.82 and 0.90, respectively. The accuracy of the algorithm was not affected by different covariates (including respiratory or valvular conditions). Conclusions: This study demonstrates the efficacy of a contact-free optical device for detecting lung congestion. Further validation of the study results across a larger and precise scale is warranted.

## 1. Introduction

Heart failure (HF) is a clinical, social and economic burden, with estimated annual worldwide costs of USD >100 billion [1]. The cost of heart failure hospitalizations in the US alone is over USD 10 billion per year [2]. Approximately 1% of the population over 40 suffers from HF [3], with the prevalence doubling every decade of life [4]. HF is a common cause of hospitalizations for those over the age of 65 and is also one of the leading causes of death in the elderly [4,5]. Over 4 million Americans are hospitalized every year due to HF, with a median length of stay of 4 days and an in-hospital mortality rate that exceeds 5% [6]. Over 30% of HF patients are readmitted within 90 days [7]. Detecting and treating early signs of congestion have been shown to improve outcomes and prevent hospitalizations [8,9]. Available remote monitoring (RM) devices leverage various techniques in order to evaluate congestion; implantable RM devices such as CardioMEMS [10] or HeartPOD [11] require an invasive procedure and might not always be cost effective or readily available for all [12,13]. Non-invasive tools measure different parameters as surrogates for congestion: vital signs, weight, symptoms, chest wall impedance, dielectric sensing and activity [14]. Although promising, telemedicine is still not considered an accurate alternative for in-person visits, and, in order to improve its efficacy, additional, sensitive and specific non-invasive means are warranted [15]. In this regard, accurate, non-invasive devices with a simplified user experience and an interface that can be operated by a layperson would undoubtedly be advantageous in improving patients’ adherence to monitoring protocol. The Gili BioSensor System (Gili) is an FDA-cleared contact-free optical system able to identify different cardiopulmonary parameters [16]. This sensor might serve as an efficient method for telemedicine to non-invasively detect early signs of lung congestion and thereby potentially prevent hospitalizations. The aim of this study was to examine the accuracy of a newly developed algorithm deployed via the Gili device as a spot measurement tool in detecting the presence of lung congestion. This is the first study to examine the Gili device in this clinical context.

## 2. Methods

### 2.1. Ethics and Study Population 

The study included patients presenting to the cardiology ward and/or outpatient clinics at the Tel Aviv Sourasky Medical Center (TASMC, Tel-Aviv Israel) that were prospectively recruited to this study (IRB approvals # TLV-0438-18, TLV-0168-19 and TLV-0692-20) after their physician/s defined them as suffering from heart failure, subjectively. All study patients provided written, informed consent before their participation. Inpatients were patients that were hospitalized for any cardiac reason, and outpatients were patients presenting to cardiovascular clinics with known heart failure in their background. The following information was recorded: clinical history and previous diagnosis, medications, physical examination and available lung ultrasound data. Patients were assigned into case/control groups based on agreement by two cardiologists as to the presence of objective and subjective evidence of lung congestion. The two cardiologists used auscultation, lung ultrasound and chest X-ray. Lung ultrasound is an assessment method that can be used at the bedside and is free of radiation, unlike chest CT. Cases were clinically congested while controls were not. Ultimately, all downstream analyses were performed twice—once when the case group included all subjects with any objective evidence of lung congestion (complete analysis) and another time when the case group included only those confirmed as cases by lung ultrasound data (subgroup analysis). Lung ultrasound was assessed using several fields and scores but was eventually utilized for this study only as a binary result.

### 2.2. Study Devices

Study participants were evaluated for lung congestion via: (1) the Gili device [17] and (2) objective reference measures indicative of increased lung fluid content, including presence of either crackles on auscultation (for all study subjects) or presence of B lines on lung ultrasound (for the subgroup analysis). The Gili device [17] (Figure 1) is a Food and Administration (FDA)-cleared optical module that operates by illuminating the inspected subject with a coherent light source (class 1 laser) while simultaneously capturing the light pattern (namely, the secondary speckle pattern) reflected off the subject by a camera. It is able to measure motion vibrations from the chest wall that are correlated with physiological or pathophysiological conditions, e.g., heart rate, respiratory rate and irregular heart rhythm [16,17,18]. Measured data may be visualized either in the time domain (e.g., as seismo/phonocardiograms [16]) or frequency domain (e.g., as spectrograms; see below). The device has been validated, and it has been shown to be capable of measuring patients with a different body mass index (BMI), chest shape, skin pigmentation and/or multiple layers of clothes [17]. It has been cleared for use in different ambient lighting conditions for spot measurement of heart rate and respiratory rate (FDA authorization #DEN200038). The patients must be compliant with the measurement procedure. 

### 2.3. Procedure and Test Methods

Following the signing of the informed consent form, the patients were evaluated for the presence of lung congestion via the above-mentioned reference measures by two cardiologists that agreed on the stratification of each individual as either a case or control. If a lung ultrasound was performed, it was conducted while evaluating 12 points on the front part of the chest and 4 points on the back [19]. Next, test measures were conducted; patients were measured while sitting in a chair in front of a table on which the Gili device was placed or in a bed with the backrest raised in front of a cardiac table on which the device was positioned. A 2 min measurement was conducted for all study subjects. During the examination time, patients were asked to remain still and also to refrain from talking. Data from the examination were exported for offline analysis. The exam was performed with help and with instructions from a technician. Most measurements were successful on the first attempt.

## 3. The Algorithm’s Diagnostic Accuracy: Development and Evaluation

In general, the algorithm development consisted of developing thousands of physiologically meaningful features in the time and frequency domains. Signals were split into overlapping buffers lasting for about 60 s, after which both heart- and respiratory-related landmarks were identified, namely, the first (In Appendix A) and second (Appendix A) heart sound components and the inhale/exhale phases. For time-dependent sound components, a high-pass filter was applied, and peaks in the signal’s envelope, corresponding to S1 and S2 heart sounds, were marked. For respiration wave components, a low-pass filter was applied (i.e., removing the sound component), after which the inhale and exhale phases were delimited. For frequency features, the spectrogram of the signal was also computed. Time features were calculated as distances and mean amplitudes between the landmarks. Frequency features were calculated from the signal’s spectrogram; for each feature, the spectrogram was sectioned vertically (frequencies) and horizontally (time) according to the landmarks (e.g., the 2 Hz to 4 Hz band during the inhale phase). The mean energy of the cross-sections was saved for use in the downstream classification process. Out of the thousand calculated features, a few tens of features were engineered and selected to distinguish between patients with pulmonary congestion and patients without (the AdaBoost [20] approach was used).

These selected features were then used to train a machine-learning-based classifier [16] that was constructed to provide an output on a scale of zero (0) to one (1), essentially representing the likelihood that a measured interval is deemed to be a case, congested (i.e., close to 1) or a control/non-congested (i.e., close to 0). A schematic sketch of the developed algorithm can be seen below in Figure 2.

Borderline scores, e.g., those close to 0.5, were deemed as inconclusive predictions and were thus regarded as non-applicable (NA) outputs. The above-mentioned, annotated data were used for algorithm training and validation purposes and were based on a cross-validation approach [21,22]. Training was carried out in order to define a machine learning model, which included parameters applied to a set of descriptors relating to the time–frequency properties of the processed signal, and to specify thresholds used to define the best cutoffs for discriminating between a predicted case, control and inconclusive prediction with borderline scores. Thresholds were calibrated so as not to deem more than 5% of the data as NAs in order to maintain ≥95% of the measurements as valid predictions, thereby maximizing the performance of the algorithm—an approach which is comparable to other marketed devices [16,23,24]. We did not include data regarding inconclusive results. A 10-fold cross-validation approach was used to estimate the preliminary accuracy of the algorithm [20,21]: The dataset was split into 10 roughly equivalent subsets (based on an available sample set of 227 patients, 10% includes ~23 patients). Multiple measurements from the same user were always included in the same subset. At each iteration, one subset was used for testing, while training was performed on the remaining nine subsets. The scores for all measurements were computed based on an algorithm that was trained on data and that did not contain any measurements related to the tested patients. The sensitivity and specify were computed at all possible classification thresholds in order to generate the accompanying receiver–operator characteristics (ROC) curve. The recommended classification threshold was selected at the point of maximal balanced accuracy.

In order to assess the effect of measurement duration on the algorithm’s performance, the 2 min Gili measurements across the entire study population were cropped into shorter buffer sizes, ranging from a 30 s buffer up to the full length of a 120 s buffer at increments of 5 s. Subsequently, the above cross-validation approach was repeated iteratively across each of these buffer sizes, and the calculated performance measures with maximum balanced accuracies per each buffer size were plotted accordingly. The final ROC plot was visualized for the complete length of the 2 min buffer.

Positive and negative predictive values (PPV and NPV) were also estimated for the complete length of the measured Gili data—once based on the prevalence of cases within the study population and a second time based on an expected prevalence of 32% in the population of HF patients presenting for routine follow-up visits [25]. Similarly, positive and negative likelihood ratios (PLR and NLR) were also estimated.

## 4. Covariate Effects and Additional Statistical Considerations

Potential effects of covariates on misclassifications by the algorithm were also sought. Specifically, since the Gili device measures motion vibration data, potential effects of valvular disorders and respiratory conditions (which may also elicit different acoustical manifests) on device performance were also assessed. For continuous and non-binary covariates, the Kolmogorov–Smirnov test was applied. For binary covariates, the Fisher’s exact test was applied. Descriptive statistics for continuous variables were expressed as means and standard deviations (STD). If continuous variables were not normally distributed, then values were reported as medians with interquartile ranges (IQR). Categorical variables were expressed using the number of observations and percentages. Intervals of 95% confidence were constructed using a 1000 bootstrap resampling with a replacement approach. A *p*-value of less than 0.05 was considered statistically significant. All analyses were conducted using MATLAB (MathWorks^®^) release 2021.

## 5. Results

### 5.1. Study Population

A total of 227 patients were recruited, from which 273 measurements were made available for this study. The mean age was 68 ± 14, with 65% males; 32% were obese (BMI > 30 kg/m^2^); 68% had a history of HF; 33% had an ongoing respiratory condition; and 40% had a valvular disorder (Table 1). No disagreements between the two cardiologists stratifying the study population into cases/controls were observed. Prevalence of lung congestion was evident in 147 (64.8%) of the evaluable measurements, accounting for 101/227 of the studied patients. Of these, 44 out of the 101 case individuals (accounting for 76 measurements) were verified via lung ultrasound and, thus, included in the subgroup analysis. For clarity, the study flowchart is depicted in Figure 3. Representative spectrograms of the signals acquired via the Gili device (visual representations of frequency spectra as varied with time [26]) are depicted in Figure 4.

### 5.2. Diagnostic Accuracy

For the complete 2 min measurement duration across all studied patients, observed sensitivity and specificity were 0.91 (95% confidence intervals (CI): 0.86–0.93) and 0.91 (95% CI: 0.87–0.94), respectively (Figure 5; Table 2). The estimated PPVs and NPVs were at least 0.82 and 0.90, respectively (Table 2). The estimated PLR and NLR were promising, with a minimum observed PLR of 10 and maximum observed NLR of 0.1 (Table 2), essentially denoting a high probability of correctly identifying or refuting the presence of lung congestion (respectively) based on the algorithm’s outputs [25]. 

### 5.3. Effect of Measurement Duration on Algorithm’s Performance

Analysis of the effect of measurement duration on the performance of the algorithm showed a general trend in favor of longer measurement durations; the minimum observed sensitivity and specificity were 0.886 and 0.87, respectively, specifically for shorter measurements. Measurements exceeding 1 min showed a mild increase in diagnostic accuracy (sensitivity and specificity ≥0.9), with maximum area under the curve (AUC) observed at measurements ≥115 s (Figure 6).

### 5.4. Covariate Effects

No effects of baseline characteristics, medical history (and, in particular, conditions with different acoustical manifestations such as pulmonary or valvular disorders) or medications on device performance were observed (Table 3).

### 5.5. Subgroup Analysis

We performed a sub-study that included 44 out of the 101 case individuals for which pulmonary congestion was verified via lung ultrasound (denoting B lines to confirm congestion), essentially serving as the case cohort in this subgroup analysis [19]. The control cohort remained unchanged (Figure 3). Baseline characteristics per this subgroup cohort are provided in Appendix A.

Observed diagnostic accuracy for this analysis exceeded that observed via the original analysis conducted for the complete study population; observed sensitivity and specificity were 0.99 (95% CI: 0.96–1) and 0.93 (95% CI: 0.88–0.95), respectively (Table 2), accounting for a balanced accuracy of 0.96. The estimated PPVs and NPVs were at least 0.86 and 0.99, respectively (Table 2). Estimated PLR and NLR were also promising, with a minimum observed PLR of 9.9 and maximum observed NLR of 0.015 (Table 2), once again denoting a high probability of correctly identifying or refuting the presence of lung congestion (respectively). In terms of covariate analysis, similar to the original analysis, no significant covariate effects were observed (Table 3).

### 5.6. The Potential Effects of Inconclusive Measurements: Analysis

The algorithm’s diagnostic accuracy was examined once more, only this time whilst also including the measurements with inconclusive predictions that were deemed NA by the algorithm—both for the analysis conducted across the complete study population and for the subgroup analysis. As expected, the observed accuracies were slightly impeded; for the complete study population, observed sensitivity and specificity were 0.91 (95% CI: 0.87–0.94) and 0.88 (95% CI: 0.84–0.91) and, for the subgroup analysis, 0.99 (95% CI: 0.96–1) and 0.89 (95% CI: 0.84–0.93), respectively (Table 2). No significant covariate effects were observed (Table 3).

## 6. Discussion

In this present study, we demonstrated the ability of a contact-free optical device to detect lung congestion by capturing the light pattern reflected from the chest wall using an optical tool measuring vibrations augmented by machine learning algorithms. Sensitivity and specificity were both 0.91 when considering the complete study population. Even in the worst-case scenario, when including inconclusive predictions (as assessed via the complementary analysis), the device still maintained a high degree of accuracy with only a mild decrease in specificity to 0.88. Interestingly, subgroup analysis on a cohort inclusive of case individuals (who were verified for congestion with lung ultrasonography) revealed improved performances, particularly for sensitivity, which reached 0.99. Such a difference between the two analyses (complete vs. subgroup) may be attributed to: (a) the relatively small case set (101 vs. 44 patients in the complete vs. subgroup analyses, respectively); and (b) the increased sensitivity that lung ultrasonography has in correctly discriminating between true cases or controls as opposed to physical examination [27].

Lung congestion is a major cause of heart failure hospitalizations [28]. Detection of such exacerbations at home can prompt early intervention, thereby potentially preventing unnecessary hospitalizations and deteriorations, thus, improving overall outcomes and reducing healthcare expenditures [29,30,31]. Several remote monitoring solutions, both invasive and non-invasive, have been developed over the years; invasive devices mainly include: (a) implantable devices, measuring intra-cardiac pressure (e.g., CardioMEMS [10,32] or HeartPOD [11]) and (b) cardiac resynchronization therapy or defibrillator-based systems, analyzing data such as heart sounds, respiration, thoracic impedance, heart rate and activity data from embedded sensors (HeartLogic HF index [33]). Although such solutions are both proven and considered the state of the art, they are still invasive and have their known implications and drawbacks such as high costs or risk of injury to blood vessels during the procedure [32].

Non-invasive solutions intended for self-use vary and include: watches, skin patches, chest belts and more [34,35]. However, these mainly enable the monitoring of vital signs which are not intended for specific identification of pulmonary congestion. Other non-invasive solutions measure bio-impedance for identification of lung fluid content but are mostly intended for use by or under the supervision of a professional (such as the ReDS™ system which uses radar technology [36]) and require close contact with the skin/body wall, which might result in discomfort or technical limitations and reluctance of patients to continue using them for the entire required monitoring duration [37]. With the rise of telemedicine, without a doubt device manufacturers need to develop solutions that are not only accurate, but also address usability issues and provide a much simpler and enhanced user experience, especially if they are to be adopted for long periods where adherence to monitoring protocol is key. In this sense, the current Gili device is small, portable and able to identify lung congestion without any contact with the patient. Although the current required measurement time seems relatively long, when considering the amount of time and potential technicalities associated with preparing some of the above-mentioned, non-invasive devices prior to taking a measurement, we feel that the overall advantage of the Gili device outweighs its drawbacks. In addition, as the algorithm matures, it is expected that it will be able to also provide the same degree of performance for shorter measurement durations of up to 1 min. The contact-free nature of the device’s functions is also advantageous in other care settings (e.g., skilled nursing facilities or nursing homes), reducing the risk for nosocomial infections contracted by touch. This is especially true in the era of the COVID-19 pandemic, which has resulted in the implementation of physical distancing and, thus, fewer patient–caregiver face-to-face appointments [38]. Moreover, the pandemic has mandated implementations of changes in the way HF patients are monitored and forced the adoption of telehealth modules so as to reduce exposure of cardiac patients to the risk of COVID-19 infection [39]. Adding a home-monitoring device to such telehealth visits/conversations enables physicians to examine the patient in a convenient and trustworthy manner, as if they presented for a physical examination during an office visit. In this respect, a recent study [39] showed that telehealth visits of HF patients during the pandemic were not associated with an increase in emergency department visits, hospital admissions or all-cause mortality, thereby further supporting the adoption of such telehealth modules. Several other studies regarding the use of telemedicine in HF patients during the pandemic showed success in decreasing hospitalizations [40,41,42,43,44]. Contact-free technologies will undoubtedly serve to enhance this approach even more. 

Although this study presents promising data, it does have some limitations that need to be acknowledged: As a pilot study, the number of patients included in the sample set was relatively small, there was no blinding and the number of case/control individuals was not completely leveled in case–control matching; the performance of the algorithm was based on a cross-validation approach. This is not commonly considered acceptable for true clinical validation purposes. Assignment of patients into case or control groups was based on agreement by two cardiologists as to the presence of objective evidence of lung congestion, and there was no clear definition of a single reference measure to be used as the gold standard across all studied patients; the performance of the device was affected by measurement duration, although the differences between shorter and longer (above 1 min) measurements were incremental. In addition, this paper does not provide statistical data regarding the transformation of the scale from sound to spectrograms. Despite this, the manufacturer is in the midst of introducing additional enhancements to the algorithm in order to favor shorter measurement durations, thereby, hopefully, reducing the appearance of motion artifacts that may be captured with longer measurement durations and introduce potential bias. Owing to the above, our results need to be validated in an additional study. We must emphasize that this a pilot study for the device. Data were combined for inpatients and outpatients. Future studies need deep characterization of the investigated patients’ characteristics with better definition of lung congestion (for example, using imaging). Furthermore, more data regarding the baseline characteristics of the heart failure need to be added (for example: duration, echocardiography at baseline and at follow up, etc.). Future studies should include functional lung tests, bio-impedance analysis, maybe DEXA data and, if possible, a CT scan (many HF patients do not show any signs of clinical congestion at rest and still have some degree of congestion in CT scans). Finally, biomarkers could give further insights into congestive phenotypes and may help to explain some of the data scattering. Future studies will also examine the influence of irregular heart rate and respiration rate on the findings and also the influence of different layers of clothes or illumination. We propose a large, randomized control study with all of the above, maybe even examining the device at patients’ homes. This preliminary study shows a good trend for using this device in larger-scale studies.

In conclusion, this is the first study to report the ability of a contact-free optical system (Gili) to identify lung congestion with a high degree of accuracy (Figure 7), with potential to benefit patients as a telehealth solution in this pandemic era. Our study examined and validated the findings among heart failure patients.

## Figures and Tables

**Figure 1 biosensors-12-00833-f001:**
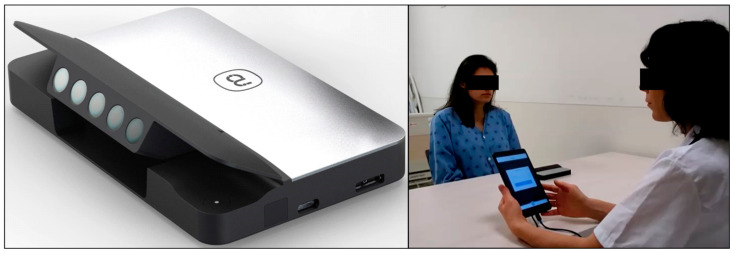
The “Gili” device and measurement setup.

**Figure 2 biosensors-12-00833-f002:**
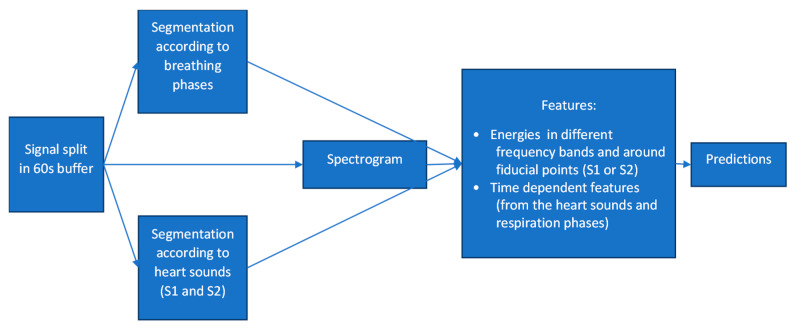
Schematic sketch of the developed algorithm.

**Figure 3 biosensors-12-00833-f003:**
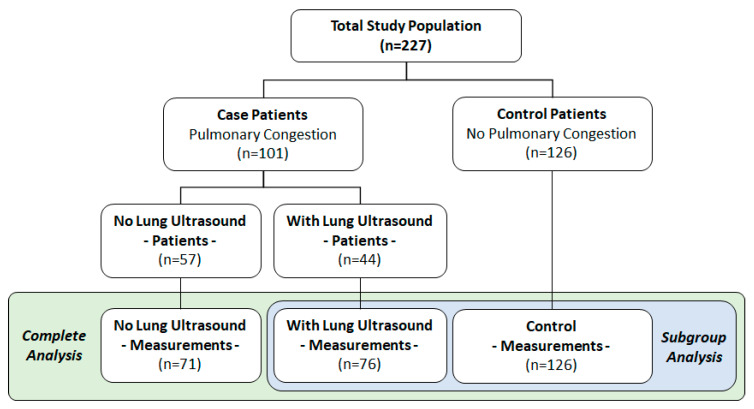
Study flowchart.

**Figure 4 biosensors-12-00833-f004:**
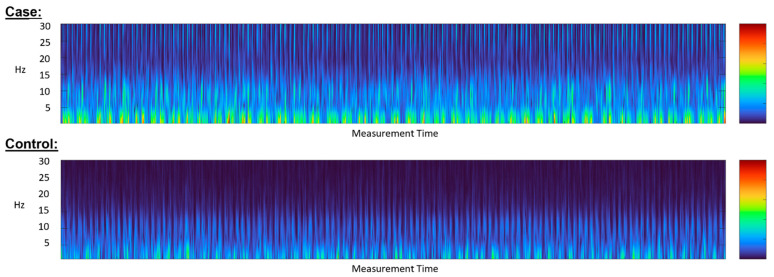
Representative spectrograms of signals.

**Figure 5 biosensors-12-00833-f005:**
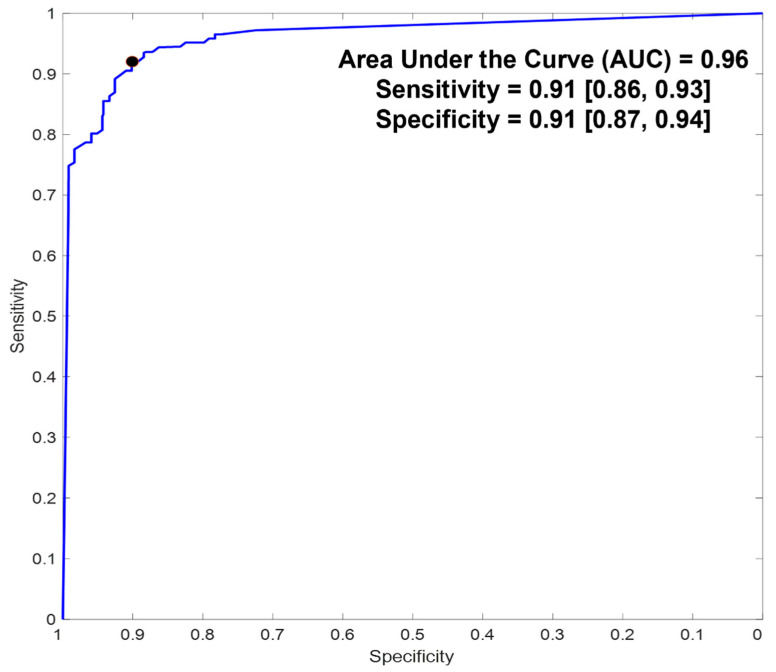
Receiver–operator characteristics. Numbers in brackets represent 95% confidence intervals (CI). Black dot marks the chosen threshold on the curve for the reported.

**Figure 6 biosensors-12-00833-f006:**
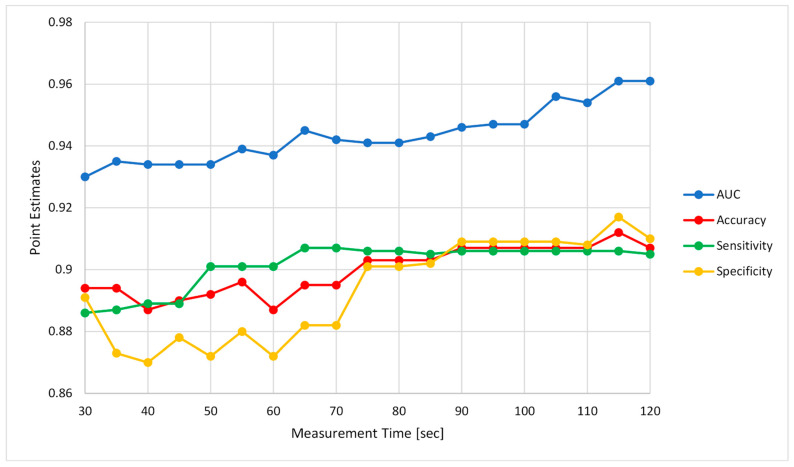
Effect of measurement duration on the algorithm’s performance.

**Figure 7 biosensors-12-00833-f007:**
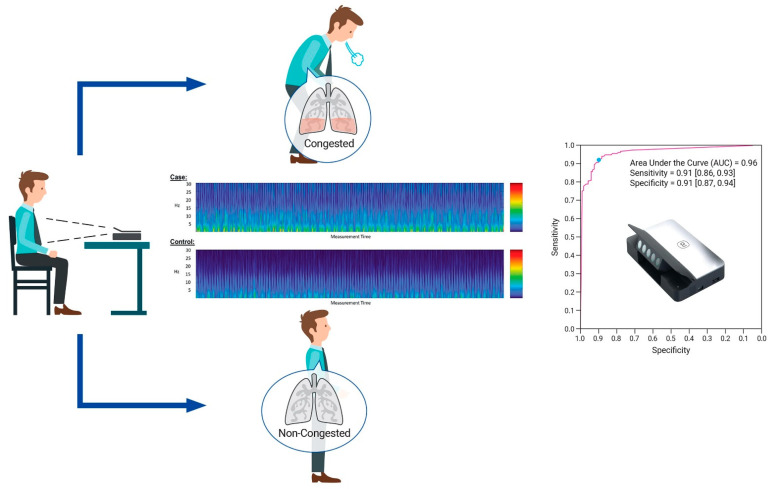
A contact-free optical device for the detection of pulmonary congestion.

**Table 1 biosensors-12-00833-t001:** Baseline characteristics of study population.

Characteristics	Total Population (*n* = 227)	Lung Congestion	
		No (*n* = 126)	Yes (*n* = 101)
Age (years): mean (SD)	68 (14)	65 (15)	73 (12)
Gender (males): *n* (%)	148 (65%)	76 (60%)	72 (71%)
Gender (female): *n* (%)	79 (35%)	50 (40%)	29 (29%)
Height (cm): mean (SD)	167 (10)	167 (10)	167 (11)
Weight (Kg): mean (SD)	79 (15)	79 (16)	80 (15)
BMI (Kg/m^2^): mean (SD)	28 (6)	28 (5)	29 (6)
BMI ≥ 30: *n* (%)	72 (32%)	34 (27%)	38 (38%)
Smoking: *n* (%)	34 (15%)	18 (14%)	16 (16%)
**Risk Factors (background):**			
Heart failure: *n* (%)	154 (68%)	76 (60%)	78 (77%)
Hypertension: *n* (%)	124 (55%)	53 (42%)	71 (70%)
Diabetes: *n* (%)	79 (35%)	22 (17%)	57 (56%)
Stroke/TIA/thromboembolism: *n* (%)	23 (10%)	7 (6%)	16 (16%)
Vascular disease: *n* (%)	114 (50%)	56 (44%)	58 (57%)
**Other Medical Conditions:**			
Respiratory conditions (COPD, asthma, pulmonary HTN, pulmonary embolism, lung cancer, etc.)	33 (15%)	11 (9%)	22 (22%)
Valvular disorders (any type; mild, moderate or severe)	91 (40%)	29 (23%)	62 (61%)
**Medications:**			
Anticoagulants: *n* (%)	71 (31%)	23 (18%)	48 (48%)
Antiplatelet: *n* (%)	129 (57%)	73 (58%)	56 (55%)
ACEi: *n* (%)	55 (24%)	35 (28%)	20 (20%)
ARB: *n* (%)	31 (14%)	11 (9%)	20 (20%)
CCB: *n* (%)	39 (17%)	14 (11%)	25 (25%)
Beta blockers: *n* (%)	130 (57%)	65 (52%)	65 (64%)
Diuretics (any): *n* (%)	112 (49%)	36 (29%)	76 (75%)
Antiarrhythmic: *n* (%)	31 (14%)	6 (5%)	25 (25%)

BMI: body mass index; TIA: transient ischemic attack; COPD: chronic obstructive pulmonary disease; HTN: hypertension; ACEi: angiotensin-converting enzyme inhibitor; ARB: angiotensin receptor blocker; CCB: calcium channel blocker.

**Table 2 biosensors-12-00833-t002:** Diagnostic accuracy of the Gili lung congestion algorithm.

Variable	Point Estimates (95% CI)	
	Complete Study Population	Subgroup Analysis
Sensitivity	0.91 (0.86, 0.93)	0.99 (0.96, 1)
Specificity	0.91 (0.87, 0.94)	0.93 (0.88, 0.95)
PPV _Sample_	0.92 (0.88, 0.95)	0.89 (0.83, 0.93)
NPV _Sample_	0.90 (0.85, 0.93)	0.99 (0.97, 1)
PPV _32%_	0.82 (0.77, 0.86)	0.86 (0.80, 0.90)
NPV _32%_	0.95 (0.92, 0.97)	0.99 (0.97, 1)
PLR	10 (6.6, 15)	9.9 (6.4, 15)
NLR	0.1 (0.07, 0.16)	0.015 (0.005, 0.049)

CI: confidence intervals. PPV: positive predictive value. NPV: negative predictive value. Sample: based on prevalence from study sample set. 32%: based on expected prevalence of 32% [25]. PLR: positive likelihood ratio. NLR: negative likelihood ratio.

**Table 3 biosensors-12-00833-t003:** Covariate effects on the algorithm’s performance.

Covariate	*p*-Values	
	Complete Study Population	Subgroup Analysis
Gender	0.430	0.313
Age	0.354	0.094
Height	0.245	0.235
BMI	0.260	0.303
BMI ≥ 30	0.565	0.298
Smoking	0.552	0.626
**Risk Factors:**		
Heart Failure	0.548	0.577
Hypertension	0.520	0.511
Diabetes	0.467	0.563
Stroke/TIA/Thromboembolism	0.584	0.811
Vascular Disease	0.456	0.552
**Other Medical Conditions:**		
Respiratory Condition (COPD, Asthma, Pulmonary HTN, Pulmonary Embolism, Lung Cancer, etc.)	0.587	0.620
Valve Disorder (Any Type; Mild, Moderate or Severe)	0.273	0.186
**Medications:**		
Anticoagulant	0.551	0.556
Antiplatelet	0.499	0.566
ACEi	0.572	0.594
ARB	0.439	0.430
CCB	0.450	0.547
Beta Blockers	0.365	0.399
Diuretics	0.409	0.412
Antiarrhythmic	0.622	0.655

BMI: body mass index; TIA: transient ischemic attack; COPD: chronic obstructive pulmonary disease; HTN: hypertension; ACEi: angiotensin-converting enzyme inhibitor; ARB: angiotensin receptor blocker; CCB: calcium channel blocker.

## Data Availability

Data used throughout this study is proprietary. Specific data types may be available upon reasonable request at the discretion of the corresponding author.

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
