# Peer review of "A Contact-Free Optical Device for the Detection of Pulmonary Congestion—A Pilot Study"

_biosensors, 2022, doi:10.3390/bios12100833_

Round 1

Reviewer 1 Report

The paper is well-aligned with the scope of the journal. The research question is well defined, relevant, and meaningful.

1.     The English writing style is acceptable even though the text holds some redundancies and ambiguous sentences.  

2.     In the abstract, the research problem must be explained with data.

3.     In the Introduction part, the strong points of this study should be further stated, and the organization of this whole paper is supposed to be provided at the end.

4.     In line 119, Which algorithm is used for classification? What are the features? There is no block diagram of the proposed method.

5.     Missing related works

6.     The effectiveness of this work is not clear. Through simulations/experiments, the authors must justify the effectiveness of the proposed method. 

Author Response

Point 1:

The English writing style is acceptable even though the text holds some ‎redundancies and ambiguous sentences.  

Response 1: Thank you for reviewing our paper. We tried addressing all the comments. We appreciate the thought and time you put into this. We went over the entire manuscript and improved the text so it will be clearer.

Point 2: In the abstract, the research problem must be explained with data.

Response 2: Thank you for this comment. We have added some numbers to the background part of the abstract in order to address the research problem more precisely. We explained the costs and numbers of hospitalization due to heart failure. We believe that this explains the research problem better.

Point 3: In the Introduction part, the strong points of this study should be further stated, and the organization of this whole paper is supposed to be provided at the end.
Response 3: Thank you for seeing the strong points of our study. We have added to the introduction, specifying what this new device might be able to bring to the clinical world (lines 69-74). We have also added to the discussion according to your suggestion.

Point 4: In line 119, which algorithm is used for classification? What are the features? There is no block diagram of the proposed method.
Response 4: Data processing and features engineering involved different segmentations of the signals: 1) heart sound (segmentations around S1 and S2 sounds); 2) respiration (segmentation into inhale and exhale phases). Time and frequency features were then computed according to the different segmentations. Adaptive boosting (AdaBoost) was used to perform the classification task. We cannot specify more details regarding the algorithm due to industrial and patent pending restrictions. We have added this to the methods section (lines 132-136). This is of course a limitation, and we have added a note accordingly in the manuscript (lines 161-162).

Point 5: Missing related works.
Response 5: We went over the references accordingly, tried to make sure as many related works as possible are added.

Point 6: The effectiveness of this work is not clear. Through simulations/experiments, the ‎authors must justify the effectiveness of the proposed method. 
Respone 6: Thank you for this comment. We added to the discussion, mainly explaining that home monitoring of lung congestion using a non-invasive touchless device will benefit patients. We think the new discussion clarifies the subject.

Reviewer 2 Report

I would like to thank the authors for the very interesting paper.

The paper is well written and the problem well stated. I have a few minor suggestions and one major change that I think is necessary for the paper publication.

Minor Corrections

The authors should check the metadata in the PDF, as for example the “Running-title” is wrong and other things are missing in the first page. But I guess this will be fixed by the editing team.

I would suggest an English proof-reading as there are a few issues with the language (for example in line 70 “patients presenting to the cardiology ward” is not correct, although the meaning is clear). Or at line 82 “Cases were congested while controls were not”. The patients may be congested, surely not the cases. This, and other, sentences must be re-written for a clear understanding of the meaning.

Something else that the authors should explain better is how they illuminated the patients. For example I could not find any information on how clothes (if present) can influence the measurements. The authors do not discuss that, but this will be very important if the device is going to be used at home by someone without medical training. Should they take their clothes off? Surely some clothing element will make the measurement of chest movement more difficult. Additionally how illumination influenced the measurement is not clear. Something else the authors should probably, at least quickly, describe and discuss.

Major Revisions

The first major problem is the fact that the authors does not (at least I have not found it) describe at all which algorithm they use (see line 120). They should be very precise since as it is now, the work is completely not reproducible. The same critic can be given on the cross-validation approach. Is not clear which one the author chose (see like 126).

The authors should explain if they did a stratified split or not (see linke 135-146). It is not clear from the paper.

More details should be added in describing the table 3, since is not really clear what is each number and for which test they are. I would add more information in the caption.

I think the major thing that I suggest the authors to change, is the removal of the predictions around the threshold. To judge the model generalization properties (and performance) all the predictions (even the wrong ones) should be kept into account. This will not change the conclusions, since the results are very good anyway, but removing specific predictions make the argument much less convincing. I would strongly suggest to give just the results with all the predictions.

Additional comments

I would be curios (but this is not a request for the authors) to know how the performance of the predictions would change if lay-persons would make the measurements instead of trained personell. This would add much value to the paper, since it would give an idea about the applicability of this method in the houses of patients or for tele-medicine (as the authors discuss).

All in all, was an interesting paper and a clever use of machine learning. From a medical point of view there is clear correlation between chest movement and lung congestions, and by using this information the authors were able to build a detection system that work, as I would expect, really well. I think the paper merits publication (with the caveat of using all predictions, including the ones that are around the threshold).

Author Response

Point 1: The authors should check the metadata in the PDF, as for example the “Running-title” is wrong and other things are missing in the first page. But I guess this will be fixed by the editing team. I would suggest an English proof-reading as there are a few issues with the language (for example in line 70 “patients presenting to the cardiology ward” is not correct, although the meaning is clear). Or at line 82 “Cases were congested while controls were not”. The patients may be congested, surely not the cases. This, and other, sentences must be re-written for a clear understanding of the meaning. Something else that the authors should explain better is how they illuminated the patients. For example I could not find any information on how clothes (if present) can influence the measurements. The authors do not discuss that, but this will be very important if the device is going to be used at home by someone without medical training. Should they take their clothes off? Surely some clothing element will make the measurement of chest movement more difficult. Additionally how illumination influenced the measurement is not clear. Something else the authors should probably, at least quickly, describe and discuss.

Response 1: Thank you for finding our paper interesting. We were happy to read and address your comments. Thank you for your time and effort. We edited the paper again in order to improve the English and the text itself. We have also mentioned clothes in the methods section (line 110). We mentioned this in section 2.2. Illumination was also added to the limitations in the Discussion (lines 350-353).

Point 2: The first major problem is the fact that the authors does not (at least I have not found it) describe at all which algorithm they use (see line 120). They should be very precise since as it is now, the work is completely not reproducible. The same critic can be given on the cross-validation approach. Is not clear which one the author chose (see like 126). The authors should explain if they did a stratified split or not (see linke 135-146). It is not clear from the paper. More details should be added in describing the table 3, since is not really clear what is each number and for which test they are. I would add more information in the caption. I think the major thing that I suggest the authors to change, is the removal of the predictions around the threshold. To judge the model generalization properties (and performance) all the predictions (even the wrong ones) should be kept into account. This will not change the conclusions, since the results are very good anyway, but removing specific predictions make the argument much less convincing. I would strongly suggest to give just the results with all the predictions.

Response 2: Regarding the algorithm. Data processing and features engineering involved different segmentations of the signals: 1) heart sound (segmentations around S1 and S2 sounds); 2) respiration (segmentation into inhale and exhale phases). Time and frequency features were then computed according to the different segmentations. Adaptive boosting (AdaBoost) was used to perform the classification task. We cannot specify more details regarding the algorithm due to industrial and patent-pending restrictions.

Regarding the cross-validation process, we did not make stratified folds. The only requirement was that for patients with more than 1 recording, all their recordings would be in the same fold. The ROC curve presented is not averaged, but built out of all the scores calculated during the 10 folds cross-validation.

The scores from the model can also be used to measure the confidence of the model, to that effect, we used the distance of the scores to the decision threshold. A rejection rate of 5% was deemed clinically acceptable. The results without any rejections (i.e. inclusive of all measurements) can be found in Supplementary.

Point 3: I would be curios (but this is not a request for the authors) to know how the performance of the predictions would change if lay-persons would make the measurements instead of trained personnel. This would add much value to the paper, since it would give an idea about the applicability of this method in the houses of patients or for tele-medicine (as the authors discuss).

All in all, was an interesting paper and a clever use of machine learning. From a medical point of view there is clear correlation between chest movement and lung congestions, and by using this information the authors were able to build a detection system that work, as I would expect, really well. I think the paper merits publication (with the caveat of using all predictions, including the ones that are around the threshold).

Response 3: Thank you so much for your compliments. We have added to the Discussion (lines 350-352) and we think all of your points should and will be addressed in our future studies using this device.

Reviewer 3 Report

In this work, the authors have realized early detection of lung congestion using machine learning techniques and Gili system. To verify the feasibility of the method, a total of 227 patients were divided into the test set and the training set with a ratio of 1:9. Then, the performances of the machine learning model were researched. Although the paper contains the necessary analysis and experimental verification, the work still has many details needed to be improved to provide convincing and clear enough results for readers. Some specific comments can be shown in the attached file.

Author Response

Point 1: In Reference 16, the authors have successfully identified atrial fibrillation in patients with the same Gili system and a similar machine-learning-based classifier with this paper. Although early detection of lung congestion was realized in this manuscript, a detailed explanation of the differences and improvements compared with Reference 16 is suggested to further present their innovations and contributions.

Response: Thank you for this comment. The device is the same. It measures motion-vibration data reflected off the chest wall. The algorithm is different, and it looks at different classifiers for each physiological parameter. We’ve updated the manuscript in the methods section to describe this change in a clearer way (section 3).

Point 2: The authors have provided representative spectrograms of the signals acquired via the Gili device as shown in Figure 3. As the authors have mentioned in the text, 2-min Gili measurements were carried out. Then, they cropped them into shorter buffer sizes. However, the information about measurement time can not be easily understood in Figure 3. Therefore, I suggest for the authors supplement the scale value of the measurement time. Besides, it seems that the scale value of Hz should be corresponding to the color bar. Therefore, I suggest for the authors change the position of the scale value of Hz.

Response: Thanks for the comment. It is possible that this a technical issue of picture files. We will contact the editor in order to make sure that the figures we will submit for the final version will answer this comment.

Point 3: To make the readers better understand the content, I suggest for the authors explain more about the aim of using the black dot in Figure 4.

Response: Thanks for the comment. The black dot marks the chosen threshold on the ROC curve for the reported sensitivity and specificity. We’ve updated the figure legend accordingly.

Point 4: The authors have elaborate much content in the discussion. However, the content about the conclusion could only be found in lines 329 to 321. Thus, an important expression of the conclusion seems to be missing. Therefore, I suggest for the authors rewrite and strengthen the content of the conclusion.

Response: Thanks for the comment. We’ve rewritten the discussion and elaborated more as suggested.

Point 5: Although the authors have explained and provided the corresponding abbreviations of most professional vocabularies, a few words or information were omitted (FDA, CI).

Response: Thanks for the comment. We’ve reviewed the manuscript and updated the missing acronyms accordingly throughout.

Point 6: The redundant author information should be removed; The content of the abstract should be rewritten in one paragraph;  The different keywords should be separated by semicolons; The format of all the numbers of the references in the main text should be modified; The format of all the tables should be modified; The subgraphs in Figure 1 and Figure 3 should be labeled with (a) and (b). The format of Reference 20 is different from others. Some parts may be missing, such as Author Contributions, Conflicts of Interest, and so on.

Response: Thank you for the comments. We’ve condensed the abstract into one paragraph. We’ve added semicolons. Reference 20 is different since it is a book. We can remove it if inappropriate. We’ve updated the tables as well.

Reviewer 4 Report

The paper is well written and the results are interesting. However, I have one question regarding the data visualization. Why the spectrogram is used? There is a number of approaches that provide better resolution in the time-frequency plane, e.g. the S-method, distributions from the Cohen class, etc. Could you please provide discussion regarding the choice of the representation?

Author Response

Point 1: The paper is well written and the results are interesting. However, I have one question regarding the data visualization. Why the spectrogram is used? There is a number of approaches that provide better resolution in the time-frequency plane, e.g. the S-method, distributions from the Cohen class, etc. Could you please provide discussion regarding the choice of the representation?

Response 1: Thank you for the comments. We agree that an S transform could give better resolution in frequencies. However for the purpose of illustration, we thought using the representation most commonly used is more adequate. The STFT should be common knowledge for all our readers and clearly displays the differences between signals with and without pulmonary congestion.

Reviewer 5 Report

The work shows promising clinical results on a contact-free device for pulmonary congestion detection. The literature review is thorough and well organized, and the test results show promising results of the proposed optical device and machine learning algorithm. However, without explaining the machine learning algorithm based on the classifier, the work has only little scientific significance. And the procedure of the data processing is not described clearly. These two areas need to be improved to show the significance of this work.

Author Response

Point 1: The work shows promising clinical results on a contact-free device for pulmonary congestion detection. The literature review is thorough and well organized, and the test results show promising results of the proposed optical device and machine learning algorithm. However, without explaining the machine learning algorithm based on the classifier, the work has only little scientific significance. And the procedure of the data processing is not described clearly. These two areas need to be improved to show the significance of this work.

Reponse 1: Regarding the algorithm: Data processing and features engineering involved different segmentations of the signals: 1) heart sound (segmentations around S1 and S2 sounds); 2) respiration (segmentation into inhale and exhale phases). Time and frequency features were then computed according to the different segmentations. Adaptive boosting (AdaBoost) was used to perform the classification task. We cannot specify more details regarding the algorithm due to industrial and patent-pending restrictions.

Round 2

Reviewer 2 Report

Thank you. I think that with the added comments and parts the paper can be published.

Author Response

Thank you.

Reviewer 3 Report

The authors have answered comments appropriately to achieve the high quality required for publishing in Biosensors. Although the paper can be considered to be accepted in its present form, some format problems due to technical issues should be improved in the further published version as the authors have stated in the responses to the reviewer’s comments.

Author Response

Thank you

Reviewer 5 Report

Since the paper's details regarding the used algorithm cannot be disclosed due to industrial and patent-pending restrictions, this paper may not have enough scientific significance. Without the provided details, the research work cannot be reproduced, and other research groups may not be benefited from this article. The current article is more suitable for other types of publications than a journal publication.

Author Response

Please see the new manuscript and answer number 3

Round 3

Reviewer 5 Report

It is good to see the general procedures of signal processing. But it would be great if more details could be added to the algorithm section so other research groups can reproduce the results. It is unclear how the landmarks were identified, how the time features and mean amplitudes were calculated, and how the frequency features were calculated.

Author Response

Thank you for reviewing our paper. We have sent the editor a new version of our manuscript. We have added several details as requested.
A high pass filter was applied to the signal to extract the sound component. Peaks in the sound envelope correspond to S1 and S2 heart sounds. The respiration wave was obtained after applying a low pass filter to the signal (i.e. removing the sound component).
Time features were calculated as distances and mean amplitudes between landmarks. Time frequency features were calculated from the spectrogram of the signal. For each feature, the spectrogram was sectioned vertically (frequencies) and horizontally (time) according to the landmarks (e.g. 2Hz to 4Hz band during inhale phase) . The mean energy of the sections was then calculated and used for the classification process.